# Blood Biomarkers from the Emergency Department Disclose Severe Omicron COVID-19-Associated Outcomes

**DOI:** 10.3390/microorganisms11040925

**Published:** 2023-04-02

**Authors:** Fiorenza Pennacchia, Eqrem Rusi, Wael Abu Ruqa, Maria Antonella Zingaropoli, Patrizia Pasculli, Giuseppina Talarico, Giuseppe Bruno, Christian Barbato, Antonio Minni, Luigi Tarani, Gioacchino Galardo, Francesco Pugliese, Marco Lucarelli, Giampiero Ferraguti, Maria Rosa Ciardi, Marco Fiore

**Affiliations:** 1Department of Sensory Organs, Sapienza University of Rome, 00185 Roma, Italy; 2Department of Human Neuroscience, Sapienza University of Rome, 00185 Rome, Italy; 3Department of Public Health and Infectious Diseases, Sapienza University of Rome, 00185 Roma, Italy; 4Institute of Biochemistry and Cell Biology (IBBC-CNR), Department of Sensory Organs, Sapienza University of Rome, 00185 Rome, Italy; 5Division of Otolaryngology-Head and Neck Surgery, ASL Rieti-Sapienza University, Ospedale San Camillo de Lellis, 02100 Rieti, Italy; 6Department of Maternal Infantile and Urological Sciences, Sapienza University of Rome, 00185 Roma, Italy; 7Medical Emergency Unit, Sapienza University of Rome, 00185 Roma, Italy; 8Department of Anesthesiology Critical Care Medicine and Pain Therapy, Sapienza University of Rome, 00185 Roma, Italy; 9Department of Experimental Medicine, Sapienza University of Rome, 00185 Roma, Italy

**Keywords:** morbidity, mortality, blood analyses, biomarkers, SARS-CoV-2, Omicron, variant

## Abstract

Background: Since its outbreak, Coronavirus disease 2019 (COVID-19), a life-threatening respiratory illness, has rapidly become a public health emergency with a devastating social impact. Lately, the Omicron strain is considered the main variant of concern. Routine blood biomarkers are, indeed, essential for stratifying patients at risk of severe outcomes, and a huge amount of data is available in the literature, mainly for the previous variants. However, only a few studies are available on early routine biochemical blood biomarkers for Omicron-afflicted patients. Thus, the aim and novelty of this study were to identify routine blood biomarkers detected at the emergency room for the early prediction of severe morbidity and/or mortality. Methods: 449 COVID-19 patients from Sapienza University Hospital of Rome were divided into four groups: (1) the *emergency* group (patients with mild forms who were quickly discharged); (2) the *hospital ward* group (patients that after the admission in the emergency department were hospitalized in a COVID-19 ward); (3) the intensive care unit (*ICU*) group (patients that after the admission in the emergency department required intensive assistance); (4) the *deceased* group (patients that after the admission in the emergency department had a fatal outcome). Results: ANOVA and ROC data showed that high-sensitivity troponin-T (TnT), fibrinogen, glycemia, C-reactive protein, lactate dehydrogenase, albumin, D-dimer myoglobin, and ferritin for both men and women may predict lethal outcomes already at the level of the emergency department. Conclusions: Compared to previous Delta COVID-19 parallel emergency patterns of prediction, Omicron-induced changes in TnT may be considered other early predictors of severe outcomes.

## 1. Introduction

Coronavirus disease 2019 (COVID-19) is a contagious respiratory illness caused by a virus, a strain of the Coronaviridae family (SARS-CoV-2, severe acute respiratory syndrome coronavirus 2) [1,2] causing millions of deaths all over the world [3]. Indeed, the subtle search for reliable biomarkers associated with COVID-19 morbidity and mortality during this pandemic crisis was a priority for clinical laboratory specialists in order to support timely and appropriate clinical management. SARS-CoV-2 is continuously evolving in the genetic code, caused by mutation or recombination occurring during the replication of the genome [4].

Recently, the World Health Organization has defined Omicron as the main circulating variant of concern [5]. According to the available data, the Omicron variant causes less severe illness and death than the other variants, such as the Delta variant [6]. In Italy, the Omicron strain appeared on November 2021, and rapidly spread all over the country [7].

The cell entry mechanism of SARS-CoV-2 can be described in three phases [8]. After the infection, SARS-CoV-2 interacts with the angiotensin-converting enzyme 2 receptor to enter respiratory cells. The second phase of infection is characterized by lung inflammation and leads to respiratory symptoms, which can range from coughing to a significant reduction in blood oxygen saturation. Finally, coagulopathies and cytokine storms occur during the third phase, leading to acute respiratory distress syndrome, cardiovascular disease, and multiorgan failure [9,10,11,12,13,14]. Based on the data collected during various studies, the experts believe that the Omicron variant prefers to attack the epithelium that lines the mucous membranes of the upper airways, rather than the lung epithelial cells; this contrasts sharply with what was observed for the Delta variant of SARS-CoV-2 [15,16,17,18,19,20]. However, first-generation SARS-CoV-2 vaccines have played a key role in efforts to bring the COVID-19 pandemic under control. Indeed, the radical drop in the number of hospitalizations and deaths after the introduction of mass vaccination crucially changed the scenario of the pandemic [21,22,23].

Older people have a higher risk of serious illness after SARS-CoV-2, and the risk increases with age, though cases of COVID-19-related deaths have even been reported in young and middle-aged adults [24,25,26]. The organic lesions of vital organs highly increase several inflammatory markers such as interleukin-6 (IL-6), C-reactive protein (CRP), lactate dehydrogenase (LDH), ferritin, and D-dimer [9,10,11,12,13,14,27]. Therefore, the early identification (i.e., data disclosed already from the emergency department) of biomarkers of morbidity and mortality is essential to help clinicians identify patients at higher risk of developing the most severe forms of COVID-19, even after vaccination [24,27]. This can help us both optimize patient treatment and reduce the workload, allowing us to better focus the appropriate treatment on patients at greater risk.

In a previous retrospective study conducted on Delta COVID-19 patients hospitalized in the emergency department at the Sapienza University Hospital “Policlinico Umberto I” in Rome, Italy, during the paroxysm’s initial phase of the pandemic, we investigated the early routine parameters of Delta COVID-19’s morbidity and mortality [24]. We collected the results of routine laboratory tests from medical records of patients that were admitted to the emergency department. As expected, we found an age-dependent morbidity increase in both the ICU and deceased groups compared with the emergency group [24]. Furthermore, data showed that aspartate transaminase (AST), alanine transaminase (ALT), LDH, total creatine kinase (CK), myoglobin (MGB), ferritin, CRP, and D-dimer could all be considered strong early predictors of COVID-19’s severity and death.

As for Omicron blood parameter studies, recent findings showed changes mainly at the blood corpuscular levels rather than biomolecular [28,29]. Thus, the aim of the present study was to uncover whether or not routine biochemical markers of the first blood withdrawal disclosed from the emergency department might be considered reliable early predictors of mortality and morbidity for the COVID-19 Omicron variant by also comparing these findings with the outcomes of the Delta variant. Indeed, we investigated the ferritin, high-sensitivity troponin-T (TnT), CK, isoenzyme MB of creatine kinase (CK-MB), LDH, MGB, ALT, AST, gamma-glutamil-transferase (GGT), fibrinogen (FBG), the international normalized ratio (INR), the activated partial thromboplastin time (aPTT), D-dimer, albumin, glycemia, and CRP. We predicted that, compared to the previous COVID-19 strains, the Omicron could show different emergency patterns of prediction, mainly for TnT and albumin.

## 2. Materials and Methods

### 2.1. Participants’ Selection and Study Design

This retrospective study is based on the first blood parameters extracted from the clinical records of 449 COVID-19 patients who accessed the emergency unit of the Sapienza University Hospital “Policlinico Umberto I” in Rome, Italy, from 1 February 2022, to 31 March 2022. Out of the 449 patients, 133 (29.6%) were not vaccinated.

In the vaccinated group, we had a heterogeneous situation where not all patients had the same number of vaccine doses (1–4 doses per person), and in the same patients, different types of vaccines were administered, and in most cases, we did not know which vaccines they received (see Table 1 for further information).

We divided the patients into four groups according to their outcomes. The first group (184, M = 77; F = 107) includes those patients who entered the emergency room and were discharged shortly after because they did not show severe symptoms (the *emergency* group).

The second group (205, M = 105; F = 100) includes those patients admitted to the emergency room and then transferred to a COVID ward and, afterward, dismissed (*hospital ward* group).

The third group (25, M = 14; F = 11) includes those who, after the ward, were transferred to the COVID intensive care units and survived (the *ICU* group).

In the fourth group (35, M = 23; F = 12), there are patients who had a fatal outcome in the emergency room, hospital ward, or ICU (the *deceased* group).

The diagnosis of SARS-CoV-2 infection was based on a positive result from real-time reverse transcription polymerase chain reaction (RT-PCR; RealStar, Altona Diagnostics, Hamburg, Germany) testing of nasopharyngeal-swab specimens, which also collected cycle threshold values [30]. Patients who reacted positively to the molecular test during recovery were transferred to the hospital’s COVID-19 wards. As a recruitment criterion, only Omicron patients were enrolled.

The University Hospital Ethical Committee approved this retrospective study (Ref. 6536), and all the study procedures followed the Helsinki Declaration of 1975, as revised in 1983, for human rights and experimentation.

### 2.2. Data Collection

For each eligible patient, we extracted information from their medical records, such as demographic characteristics (age and sex), vaccination, symptoms, comorbidities, and laboratory analytical results. The results of the available laboratory tests were collected when patients were initially admitted to the emergency unit and included: liver biomarkers (AST, ALT, and GGT); LDH; cardiac biomarkers (TnT, CK, CK-MB, and MGB); inflammatory biomarkers (ferritin and CRP); biomarkers of coagulation (fibrinogen, INR, aPTT, and D-dimer); albumin; and glycemia (see Table 2).

We also found incomplete data for pancreatic enzymes from the emergency department, which was insufficient to provide a reliable comparison between the study groups. For this reason, we did not include these enzymes in this study. As for IL-6, for a few patients, we had this information from the ward clinical record, but since it was never dosed in the emergency department and was not considered a routine analysis, we excluded IL-6 from the study.

### 2.3. Laboratory Examination

The patient’s peripheral blood was collected in vacutainer tubes for blood testing. As tube additives, sodium citrate was used for coagulation testing, and separating gel was used to obtain blood serum. The BCS XP System automatic hemostatic analyzer (Siemens Healthcare, Germany) was used to test coagulation parameters such as D-dimer (reference range: 50–420 μg/L) and fibrinogen (reference range: 1.5–4 g/L), using respectively immunoturbidimetric assay and Clauss modified methods. The interassay coefficient of variation (CV) was, respectively, 7.9% at a serum D-dimer of 200 μg/L and 2.5% at a serum fibrinogen of 2 g/L. Derived international normalized ratio (INR) (reference range: 0.8–1.2) and activated partial thromboplastin time (aPTT) ratio (reference range: 0.8–1.14) were determined coagulometrically; the interassay coefficient of variation (CV), tested in normal and pathological samples, was between 0.3 and 2.8% for aPTT. Biomarkers of damage, including AST (reference range: 9–45 U/L), ALT (reference range: 10–40 U/L), CK (reference range: 20–200 U/L), GGT (reference range: 8–61 U/L), and LDH (reference range: 135–225 U/L), were assessed. CV was respectively 2.3% at a serum AST of 30 U/L, 2.6% at a serum ALT of 24 U/L, 3.2% at a serum CK of 18.7 U/L, 3.2% at a serum GGT of 46.8 U/L, and 2.7% at a serum LDH of 124 U/L. Markers of inflammation, such as ferritin (reference range: male 30–400 μg/L; female 15–150 μg/L) and CRP (reference range: 0.1–6 mg/dL), were measured too. CV was 2.8% at a serum ferritin of 26.1 μg/L and 1.3% at a serum CRP of 39.9 μg/L, respectively. Biochemical parameters such as serum albumin (reference range: 3.5–5 g/dL) and glucose (reference range: 70–100 mg/dL) were also analyzed. Interassay CV was, respectively, 0.9% at a serum albumin of 51 g/L and 1.1% at a serum glucose of 5.3 mmol/L. All the above parameters were measured using standard colorimetric and/or enzymatic methods on a Cobas C-501 analyzer (Roche Diagnostics, Germany). Cardiac biomarkers like TnT (reference range: until 0.014 μg/L), MGB (reference range: 28–72 μg/L) and CK-MB (reference range: until 4.94 μg/L) were measured on a Cobas E-601 analyzer, using sandwich immunological methods (Roche Diagnostics, Germany). The interassay CV was respectively 2.7% at a serum TnT of 0.017 μg/L, 1.9% at a serum Myoglobin of 60.5 μg/L, and 1.4% at a serum CK-MB of 5.34 ng/mL.

### 2.4. Statistical Analysis

According to methods previously described [31,32], data were analyzed to assess normality by Pearson’s chi-squared test. A two-way analysis of variance (ANOVA) (comparing *emergency* vs. *hospital ward* vs. *ICU* vs. *deceased* and men vs. women) was used to analyze the laboratory parameters and the vaccination data. Post hoc comparisons were carried out using Tukey’s HSD test. The Spearman correlation test was used to investigate the correlation between the laboratory data and the age of the patients [33]. A receiver operating characteristic (ROC) analysis was performed to measure the diagnostic/predictive accuracy of each variable [31]. All analyses were performed using Epitools by Ausvet (Canberra, Australian Capital Territory, Australia) and StatView (Abacus Corporation, Baltimore, MD, USA).

## 3. Results

The patients’ clinical severity of COVID-19 symptoms and comorbidities for each group based on the information extracted from the emergency clinical records is shown in Table 3. Interestingly, ventilation was not very necessary, and only five men from the *deceased* group underwent mechanical ventilation.

According to the Omicron clinical severity of the present cohort of patients, the most common comorbidities found in the *ICU* and *deceased* groups were, as expected (since these were elderly individuals), of a cardiovascular, neurological, oncological, and metabolic nature. Indeed, the analysis of concomitant comorbidities between groups by ANOVA clearly discloses the highest values in the *deceased* and *ICU* groups compared to the *emergency* patients [F(3,441) = 30.002, *p* < 0.001, see post hoc comparisons in Figure 1, left panel] in the absence of a sex effect. 

Quite interestingly, concomitant comorbidities were also higher in the *hospital ward* patients. Furthermore, ANOVA also revealed a higher frequency of comorbidities among vaccinated compared to unvaccinated patients [F(1,441) = 4.40, *p* < 0.05; see post hoc comparisons in Figure 1, right panel], particularly evident for the *hospital ward* group, again in the absence of a sex effect.

We noted that among the most common comorbidities were hypertension, ischemic heart disease, atrial fibrillation, lung and colorectal cancers, leukemias, lymphomas, Alzheimer’s and Parkinson’s diseases, stroke, psoriasis, rheumatoid arthritis, systemic lupus erythematosus, and a few cases of glaucoma.

The age of the recruited Omicron COVID-19 individuals is shown in Figure 2. The age range was 17–88 years (44.55 ± 1.31) for the *emergency* group, 19–98 years (65.24 ± 1.21) for the *hospital ward* group, 36–88 years (65.76 ± 3.11) for the *ICU* group, and 56–104 years (78.20 ± 2.07) for the *deceased* group. Indeed, ANOVA data show an effect of morbidity [F(3,441) = 63.13, *p* < 0.01] because of the higher values of the *hospital ward*, *ICU*, and *deceased* groups compared with the *emergency* group (ps < 0.01 in post hoc analyses). Furthermore, post hoc analyses revealed a higher age of the *deceased* individuals compared to the individuals in the *ICU* and *hospital ward* (ps < 0.05). Data also did not show a sex effect [F(1,441) = 1.24, *p* = 0.26] or the interaction morbidity x age [F(3,441) = 0.93, *p* = 0.96]. The mortality due to Omicron in patients attending the hospital’s emergency department was 7.79%.

In order to assess the recruited individuals’ lung function, we used the Horowitz Index for Lung Function (P/F Ratio) (Figure 3). In the absence of a sex effect (Figure 3 is presented without the sex parameter), the data clearly evidence a main effect of morbidity with low values that depend on the Omicron COVID-19 severity [F(3,440) = 28.20, *p* < 0.01]. Indeed, in the post hoc comparisons, the *deceased* group had the lowest values compared to the other groups (ps < 0.05); furthermore, the individuals in the *ICU* and *hospital ward* groups also had low values when compared to the individuals in the *emergency* group (ps < 0.05).

The blood parameters determined by ANOVA are shown in Figure 4, Figure 5, Figure 6 and Figure 7, while Table 4 shows the relative F and dF values (no sex effects were found and these data were not shown, except for ferritin). A main effect of morbidity was disclosed by ANOVA for CK, TnT, FBG, GGT, glycemia, CRP, aPTT, LDH, D-dimer, MGB, AST, and ferritin, with a significant potentiation in *deceased* individuals compared to the individuals in the *emergency* group (ps < 0.05 in post hoc analyses for the differences between groups).

Furthermore, the *ICU* and *hospital ward* groups displayed also high values if compared to the values of the *emergency* group for TnT, FBG, glycemia, GGT, CRP, LDH, D-dimer, MGB, AST, and ferritin (ps < 0.05 in post hoc analyses for the differences between groups). As for ferritin, as previously shown for the Delta variant [24], the men of the *deceased* group had the highest potentiation (ps < 0.05 in post hoc analyses for the differences between groups).

Table 5 and Table 6 show the ROC data. The area under the curve (AUC, Table 5) scores for TnT, FBG, glycemia, CPR, LDH, albumin (although in the absence of significant ANOVA data), D-dimer, MGB, and ferritin (in both men and women) disclose the highest values (in bold in Table 5) in the *deceased* group.

As for the *ICU* group, we found similar findings for FBG, D-dimer, and ferritin only in women. The positive predictive values (PPV) in the *deceased*, *ICU*, and *hospital ward* groups and the negative predictive values (NPV) in the *emergency* group based on the reference values for P/F ratio, CK, CK-MB, TnT, FBG, GGT, glycemia, CRP, INR, aPTT, LDH, albumin, D-dimer, MGB, AST, ALT, and ferritin (men and women—see Section 2 section) are shown in Table 6.

In the *deceased* group, the highest PPV scores were evidenced for FBG, glycemia, LDH, D-dimer, and ferritin in both women and men. The highest PPV scores of the *ICU* group were the FBG, D-dimer, and ferritin in women. Quite interestingly, the NPV highest values of the *emergency* group were found for AST, ALT, MGB, CK, CRP, and ferritin (for women).

Table 7 shows the Spearman correlations for the age parameter in the *deceased* group. Indeed, in men, a positive correlation was found for TnT but a negative one for ferritin.

As for the effect of vaccination (Table 1), the contingency tables show a significant X^2^ value due to the differences in percentage between the *deceased* group and the other groups [dF(3), X^2^ = 15.63, *p* < 0.01, contingency coefficient of 0.183]. However, by analyzing vaccination as the number of doses, in the absence of sex effects, significant differences were found in the two-way ANOVA between the *deceased* and the *emergency* groups, with lower values in patients with a fatal outcome [F(3,419) = 5.55, *p* = 0.014; see Figure 8 for post hoc analyses]. The vaccination effects on the single analyzed biomarkers are shown in Table 8 (expressed as a two-way ANOVA in the absence of a sex effect). Quite surprisingly, the above-shown effects on Omicron COVID-19 morbidity were associated with an effect of vaccination on TnT and LDH. In particular, unvaccinated individuals had higher LDH levels, mainly in the *deceased*, *hospital ward*, and *ICU* groups, whereas unvaccinated patients had lower levels of TnT compared to vaccinated patients (see Figure 9).

## 4. Discussion

In this retrospective biochemical investigation, we analyzed early routine laboratory data in a population of 449 Omicron COVID-19 patients who developed different outcomes, whether by attending/not attending a hospital ward, an ICU, or facing a fatal fate. Based on this methodological schedule, we assessed the clinical records of patients who were admitted to the emergency unit. Some of these patients were subsequently discharged because they did not show severe signs and symptoms. Others attended the dedicated COVID-19 hospital ward to be released shortly without severe consequences. Unfortunately, other patients underwent severe COVID-19 and attended the ICU or died from Omicron COVID-19. In particular, we show that patients coming from an emergency department will later develop lethal COVID-19; they had potentiated values of several early laboratory biomarkers used for the initial screening of COVID-19 and general health. Indeed, ANOVA data revealed that CK, TnT, FBG, GGT, glycemia, CRP, aPTT, LDH, D-dimer, MGB, AST, and ferritin were strongly increased in patients from the *deceased* group compared to the *emergency* group, as previously observed in the hospitalized group [29,34,35,36,37,38].

Interestingly, but not surprisingly, compared to the Delta COVID-19, the number of intubated Omicron patients was very low (see Table 3), again underlining the differences in morbidity between Omicron and the previous COVID-19 variants [5,39]. However, as shown by the P/F ratio data, the severe Omicron COVID-19 infection induced a reduction in respiratory abilities [40,41] more evident in individuals of the *deceased* group.

Other interesting findings of the present investigation are the vaccination data, which show the highest percentage of unvaccinated women in the *deceased* group but also a high percentage of vaccinated men (Table 1). Furthermore (Table 8 and Figure 8), a significant elevation in LDH in unvaccinated individuals was observed, particularly in the *deceased*, *ICU*, and *hospital ward* groups. Surprisingly, vaccinated patients had higher TnT values when compared to unvaccinated individuals. However, such differences in TnT levels were not associated with the frequency of pre-existing cardiac diseases. However, the group sizes are too small to draw any conclusions.

The mortality due to Omicron in patients attending the hospital emergency department was 7.79%, mostly concentrated, as expected in older people. Moreover, under the present experimental conditions of this retrospective study, with a relatively low number of recruited individuals, vaccination, when measured as the number of doses, reveals that the highest number of vaccine doses was found in the *emergency* group, indicating greater protection against Omicron-associated morbidity and mortality. However, it should be noted that vaccination data are based on the personal declarations of enrolled patients, which, for reasons related to stigma or other personal motifs, could not be true, as observed for other data involving personal declaration [42], underestimating the number of doses and/or the number of unvaccinated patients.

As for the COVID-19 vaccine’s effectiveness against Omicron, available data show that the primary immunization with more than two doses may substantially increase protection, but that protection waned in a few weeks [43,44]. Data also demonstrate that mRNA vaccines were found to be highly effective in preventing COVID-19-associated hospital admissions related to the Alpha, Delta, and Omicron variants, but three vaccine doses were required to achieve protection against Omicron, similar to the protection that two doses provided against the delta and alpha variants [45]. Among adults admitted to hospitals with COVID-19, the Omicron variant was associated with less severe disease than the Delta variant but still resulted in substantial morbidity and mortality [45]. Vaccinated patients admitted to the hospital with COVID-19 had significantly lower disease severity than unvaccinated patients for all the variants [45,46].

In this cohort of Omicron-afflicted patients, factors influencing mortality were age and the number of concomitant comorbidities (see Figure 1 and Figure 2). Indeed, early in the COVID-19 pandemic, people with underlying comorbidities were overrepresented in hospitalized cases of COVID-19, and the relationship between comorbidity and COVID-19 outcomes was potentiated by age [47,48]. As shown in Table 3, the destructive effects of COVID-19 on elderly people with cardiovascular diseases have been again shown. Thus, specific therapies focusing on both cardiovascular and aging-related pathways [49,50] could positively impact the COVID-19 treatment, such as by potentiating, in the elderly, the immune system efficiency and the endogenous antioxidant/anti-nitrosative systems [51,52,53].

The AUC values of the receiver operating characteristic curve indicate that a potentiation in a combination of several routine blood parameters at the emergency department level might provide early indications of severe or fatal Omicron outcomes, as previously shown by the Delta COVID-19 [24,31,54,55]. Indeed, the AUC scores for TnT, FBG, glycemia, CPR, LDH, albumin, D-dimer MGB, and ferritin for both men and women were particularly elevated in the *deceased* group patients. However, for the *ICU* group patients, the highest AUC scores were found for FBG, D-dimer, and ferritin, but only in women. The PPV data extend and confirm these findings, showing for patients of the *deceased* group that elevations in FBG, glycemia, LDH, D-dimer, and ferritin for both women and men (but not TnT, CRP, albumin, or MGB) should be carefully evaluated by professionals treating Omicron COVID-19-affected individuals [56,57,58]. Moreover, the PPV values of FBG, D-dimer, and ferritin, but only for women, were also elevated in patients in the *ICU* group. An additional finding of the ROC data shows that the NPV scores support the hypothesis that early normal-range blood parameters (i.e., CK-MB, TnT, FBG, AST, ALT, CK, CRP, albumin, and INR) in Omicron COVID-19 individuals might provide mild or light effects. Compared to the previous emergency department-based Delta COVID-19 study [24], the AUC scores evidence of differences in TnT. In particular, the elevation in TnT in the *deceased* group was correlated with the age of the subjects (see Table 7), but only for men. Another difference with the previous emergency department-based Delta COVID-19 study [24] was the absence of Omicron effects on ALT.

In previous Omicron studies based on the hematological characteristics of COVID-19 patients, the authors disclosed data on the cellular part of the blood [28,29]. They found that changes in white blood cells, neutrophils, lymphocytes, platelets, monocyte-to-lymphocyte ratio, neutrophil-to-lymphocyte ratio, platelet-to-lymphocyte ratio, and platelet volume-to-platelet ratio may be useful in the early diagnosis of Omicron COVID-19. These investigators stated that also changes in CRP could provide reliable information in the Omicron COVID-19 diagnosis [29]. We confirmed this CRP data, and, since we focused on the blood biochemical parameters, we did show that TnT, FBG, glycemia, CPR, LDH, albumin, D-dimer, MGB, and ferritin for both men and women might represent useful indicators of severe morbidity and mortality for Omicron COVID-19.

In a study dealing with cytokine profiling [59] in different COVID-19 variants (ancestral Wuhan strain, Alpha, Delta, and Omicron), the authors showed that out of nearly 30 cytokines, only four showed stable elevation independently of the variant (IL-6, IL-10, IL-18, and IL-27), proposing them as ‘constant’ markers for COVID-19 infection. However, the profile of these cytokines in Omicron COVID-19 patients is not considered, at least in Italy, a routine measurement at the level of the emergency department.

Another study investigated the differences in certain inflammatory biomarkers between Alpha, Delta, and Omicron SARS-CoV-2 variants [60]. Indeed, the authors hypothesized that Omicron infections could cause less inflammation compared to Alpha and Delta infections, correlating with lower mortality [60]. By analyzing the adjusted odds ratios of the first laboratory results, the authors found that abnormalities elicited by the Delta variant in CRP, ferritin, albumin, LDH, ALT, and AST were higher than in Omicron [60]. The authors found also that the results remained significant after stratifying the outcomes based on vaccination status, evidencing that the Omicron variant caused milder inflammatory responses and lower mortality than other variants [60].

The strength and originality of this study lay in the classification of Omicron COVID-19 individuals according to their final outcomes. The groups of patients were established by means of criteria correlated with severe COVID-19 disease. The present retrospective investigation is focused on the levels of (i) the blood biochemical parameters and (ii) the emergency room, aimed at early predicting severe COVID-19 outcomes by comparing four different groups of Omicron patients (see Section 2; *emergency* only patients, *hospital ward* patients, *ICU* patients, and *deceased* patients). In order to disclose severe outcomes, analogous investigations were definitely carried out, but with groups of patients and other experimental schedules [61,62]. Typically, the main criteria previously used were oxygen saturation levels, fever, age, respiratory rate, respiratory distress, the presence of bilateral and peripheral ground-glass opacities, and arterial blood oxygen partial pressure [63].

However, the Omicron strain, as well as the other COVID-19 variants, is characterized by great variability in the clinical manifestations, which are not always related to a final prognosis [64]. For this reason, in this study, we assessed the early biochemical routine blood parameters in patients who developed a light, mild, severe, or fatal prognosis.

This work has some limitations. In the unvaccinated group, immunity against SARS-CoV-2 infection was not ruled out by checking for previous infections, and since the recovery of the recruited patients’ serum samples is not possible, this issue is considered a study limit. Moreover, because the data of this retrospective study derive from a single university hospital, the *n* of the four groups of COVID-19 patients was different and small for the *ICU* and *deceased* groups (but this, fortunately, depends on the quite less severe characteristics of Omicron), so certain biases could have arisen, including the vaccination findings. In addition, assembling broad and complete pieces of information in the emergency section of the medical records was difficult and complex because, due to COVID-19, the hospital facilities were and still are, anyway, under pressure. For this reason, many biomedical findings are missing.

## 5. Conclusions

This study revealed an association between routine blood biomarkers and the severity of outcomes in Omicron patients. We strongly believe that dosing these biomarkers at an early stage can support clinicians in monitoring the ongoing infection and, most importantly, improve the patient’s treatment and prognosis. Furthermore, the disclosed similarities and differences in the laboratory data between the COVID-19 variants indicate common patterns of alterations whose knowledge could be useful in the case of new emerging variants. Thus, this research may represent an additional attempt to unravel early biomolecular markers of COVID-19 progression and could also be of interest for investigations dealing with human diseases induced by bacterial or viral infections, such as other coronaviruses.

## Figures and Tables

**Figure 1 microorganisms-11-00925-f001:**
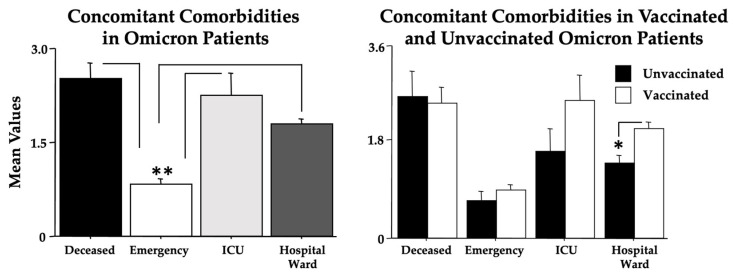
Concomitant comorbidities of the recruited individuals (**left panel**) with or without vaccination (**right panel**). The error bars indicate pooled standard error means (SEM) derived from the appropriate error mean square in the ANOVA. The asterisks (** *p* < 0.01; * *p* < 0.05) indicate post hoc differences between groups.

**Figure 2 microorganisms-11-00925-f002:**
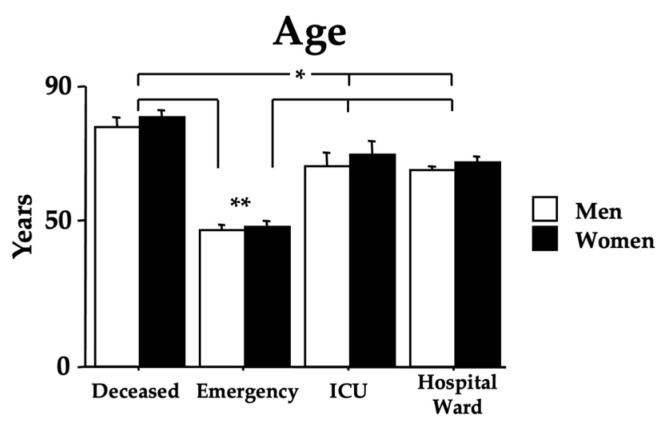
Age in years of the recruited individuals. The error bars indicate pooled standard error means (SEM) derived from the appropriate error mean square in the ANOVA. The asterisks (* *p* < 0.05; ** *p* < 0.01) indicate post hoc differences with the emergency group.

**Figure 3 microorganisms-11-00925-f003:**
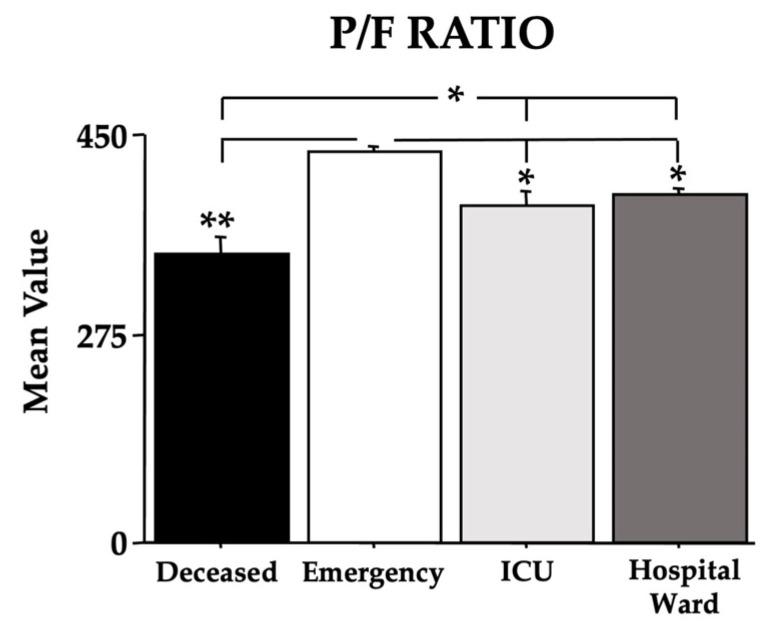
The Horowitz Index for Lung Function (P/F Ratio) of the recruited individuals. The error bars indicate pooled standard error means (SEM) derived from the appropriate error mean square in the ANOVA. The asterisks (* *p* < 0.05; ** *p* < 0.01) indicate post hoc differences between groups.

**Figure 4 microorganisms-11-00925-f004:**
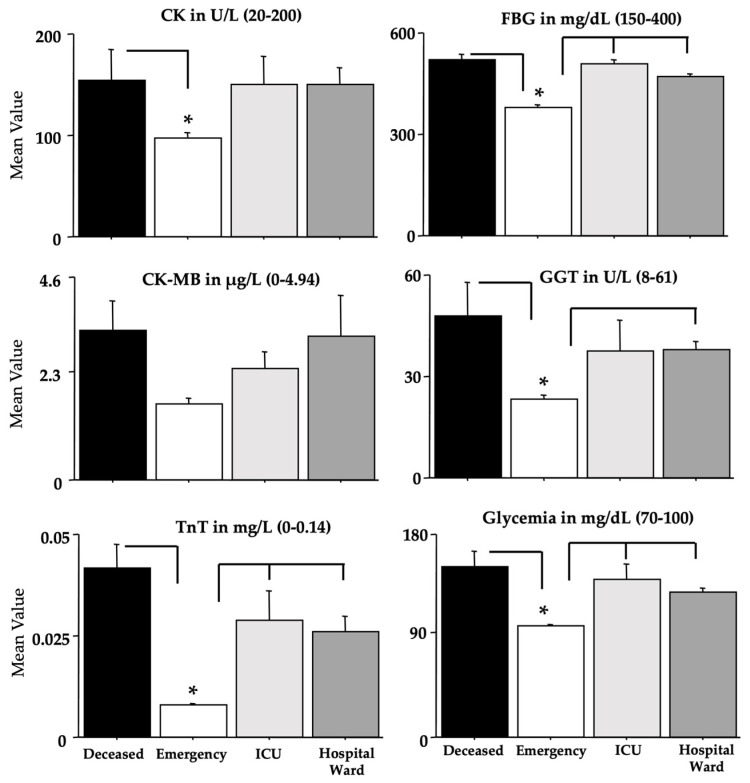
CK, FBG, CK-MB, GGT, TnT, and glycemia in recruited individuals (and relative reference ranges). The error bars indicate the pooled standard error means (SEM) derived from the appropriate error mean square in the ANOVA. The asterisks (* *p* < 0.05) indicate post hoc differences between groups.

**Figure 5 microorganisms-11-00925-f005:**
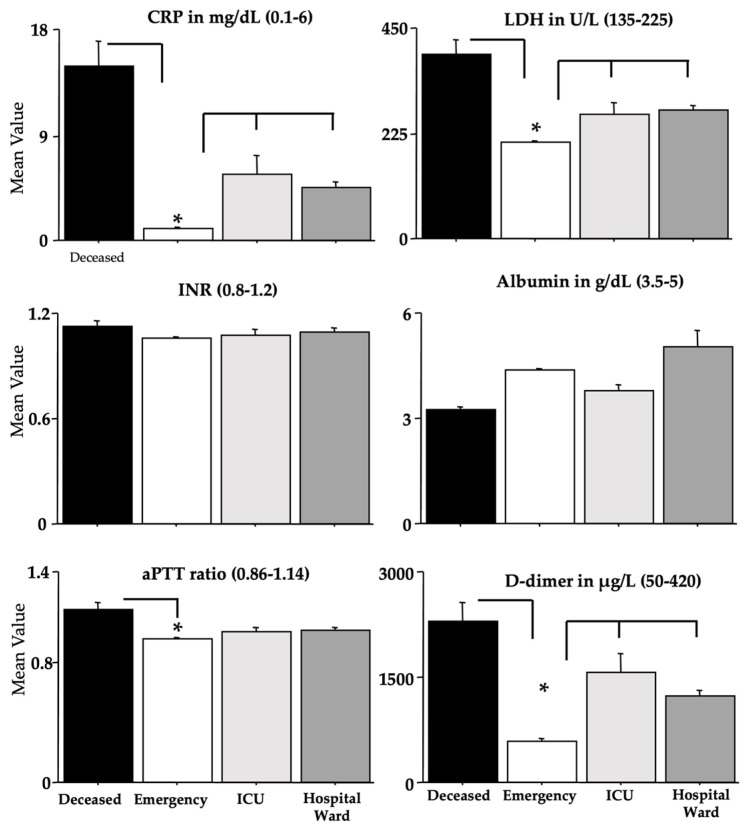
CRP, LDH, INR, albumin, aPTT, and D-dimer in recruited individuals (and relative reference ranges). The error bars indicate the pooled standard error means (SEM) derived from the appropriate error mean square in the ANOVA. The asterisks (* *p* < 0.05) indicate post hoc differences between groups.

**Figure 6 microorganisms-11-00925-f006:**
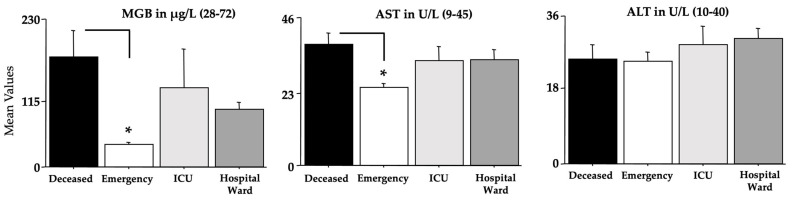
MGB, AST, and ALT in recruited individuals (and relative reference ranges). The error bars indicate the pooled standard error means (SEM) derived from the appropriate error mean square in the ANOVA. The asterisks (* *p* < 0.05) indicate post hoc differences between groups.

**Figure 7 microorganisms-11-00925-f007:**
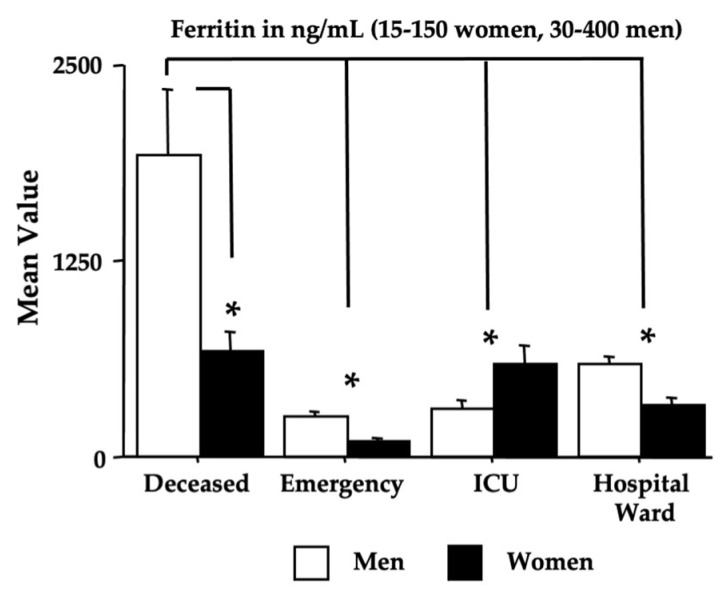
Ferritin in recruited women and men (and relative reference ranges). The error bars indicate the pooled standard error means (SEM) derived from the appropriate error mean square in the ANOVA. The asterisks (* *p* < 0.05) indicate post hoc differences between groups.

**Figure 8 microorganisms-11-00925-f008:**
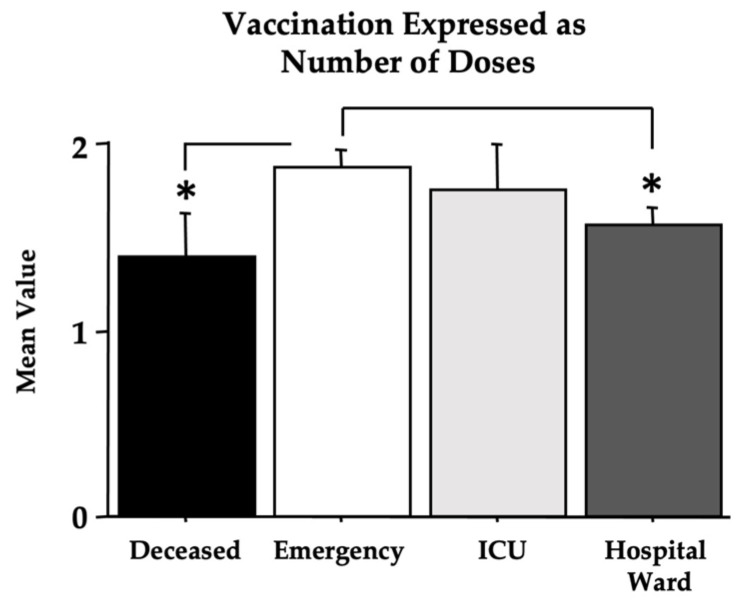
The vaccination effect is expressed as the number of doses of the vaccine. The error bars indicate pooled standard error means (SEM) derived from the appropriate error mean square in the ANOVA. The asterisks (* *p* < 0.05) indicate post hoc differences between groups.

**Figure 9 microorganisms-11-00925-f009:**
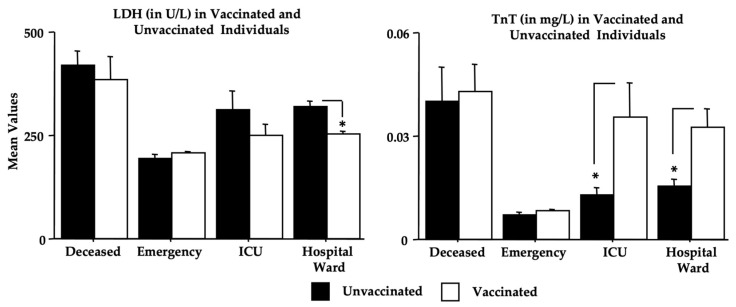
Vaccination effects on LDH and TnT (see Section 3). The error bars indicate pooled standard error means (SEM) derived from the appropriate error mean square in the ANOVA. The asterisks (* *p* < 0.05) indicate post hoc differences between groups.

**Table 1 microorganisms-11-00925-t001:** Doses of vaccines in the recruited individuals.

	*Emergency*		*Hospital Ward*		*ICU*		*Deceased*	
	M (77)	F (107)	M (105)	F (100)	M (14)	F (11)	M (23)	F (12)
Vaccinated	65 (84.4%)	81 (75.7%)	69 (65.7%)	65 (65%)	10 (71.4%)	8 (72.7%)	14 (60.8%)	4 (33.3%)
1 dose	4 (6.15%)	8 (9.88%)	3 (4.35%)	4 (6.15%)	2 (20.00%)	0 (0.00%)	0 (0.00%)	0 (0.00%)
2 doses	26 (40.00%)	33 (40.74%)	24 (34.78%)	29 (44.62%)	3 (30.00%)	3 (37.50%)	4 (28.57%)	1 (25.00%)
3 doses	38 (55.07%)	27 (41.54%)	5 (50.00%)	27 (41.54%)	5 (50.00%)	5 (62.50%)	10 (71.43%)	3 (75.00%)
4 doses	2 (3.08%)	0 (0.00%)	0 (0.00%)	0 (0.00%)	0 (0.00%)	0 (0.00%)	0 (0.00%)	0 (0.00%)
Unknown n° of doses	6 (9.23%)	5 (6.17%)	4 (5.80%)	5 (7.69%)	0 (0.00%)	0 (0.00%)	0 (0.00%)	0 (0.00%)
Unvaccinated	12 (15.6%)	26 (24.3%)	36 (34.3%)	35 (35%)	4 (28.6%)	3 (27.3%)	9 (39.2%)	8 (66.7%)

**Table 2 microorganisms-11-00925-t002:** Number of routine analyses available for each group and considered for the statistical analyses.

	*Emergency*	*Hospital Ward*	*ICU*	*Deceased*
N. of patients	184	205	25	35
Ferritin	161	179	21	31
CK	178	197	25	33
CK-MB	171	185	24	31
TnT	178	193	24	32
FBG	178	194	25	33
GGT	161	187	23	32
Glycemia	178	199	25	33
CRP	157	163	15	28
INR	177	197	25	33
aPTT	177	196	25	32
LDH	179	198	25	33
Albumin	157	179	22	33
D-dimer	170	186	24	31
MGB	111	128	22	22
AST	179	198	25	33
ALT	178	198	25	33

**Table 3 microorganisms-11-00925-t003:** Recorded symptoms and comorbidities characterizing the recruited individuals for each group.

	*Emergency*		*Hospital Ward*		*ICU*		*Deceased*	
	M (77)	F (107)	M (105)	F (100)	M (14)	F (11)	M (23)	F (12)
COVID-19 Symptoms								
Fever	31 (40.26%)	60 (56.07%)	55 (52.38%)	43 (43.00%)	5 (35.71%)	5 (45.45%)	11 (47.83%)	7 (58.33%)
Cough	27 (35.06%)	51 (47.66%)	36 (34.29%)	27 (27.00%)	1 (7.14%)	2 (18.18%)	8 (34.78%)	3 (25.00%)
Dyspnea	14 (18.26%)	30 (28.04%)	42 (40.00%)	31 (31.00%)	2 (14.29%)	6 (54.55%)	15 (65.22%)	9 (75.00%)
Asthenia	10 (12.99%)	23 (21.50%)	8 (7.62%)	12 (12.00%)	0 (0.00%)	3 (27.27%)	3 (13.04%)	2 (16.67%)
Rhinitis	6 (7.89%)	5 (4.67%)	6 (5.61%)	2 (2.00%)	0 (0.00%)	0 (0.00%)	0 (0.00%)	1 (8.33%)
Memory deficits	0 (0.00%)	0 (0.00%)	1 (0.95%)	0 (0.00%)	0 (0.00%)	0 (0.00%)	1 (4.35%)	1 (8.33%)
Vertigo	2 (2.60%)	3 (2.80%)	0 (0.00%)	4 (4.00%)	0 (0.00%)	0 (0.00%)	0 (0.00%)	0 (0.00%)
Anosmia	1 (1.30%)	2 (1.87%)	1 (0.95%)	4 (4.00%)	0 (0.00%)	0 (0.00%)	0 (0.00%)	1 (8.33%)
Ageusia	1 (1.30%)	2 (1.87%)	1 (0.95%)	2 (2.00%)	0 (0.00%)	0 (0.00%)	0 (0.00%)	1 (8.33%)
Depression or anxiety	3 (3.90%)	2 (1.87%)	1 (0.95%)	4 (4.00%)	1 (7.14%)	0 (0.00%)	0 (0.00%)	1 (8.33%)
Brain fog	1 (1.30%)	0 (0.00%)	1 (0.95%)	0 (0.00%)	0 (0.00%)	0 (0.00%)	1 (4.35%)	0 (0.00%)
Epistaxis	0 (0.00%)	0 (0.00%)	1 (0.95%)	0 (0.00%)	0 (0.00%)	0 (0.00%)	0 (0.00%)	1 (8.33%)
Arthralgia or myalgia	13 (16.88%)	32 (29.91%)	7 (6.67%)	7 (7.00%)	2 (14.29%)	1 (9.09%)	0 (0.00%)	2 (16.67%)
Headache	9 (11.69%)	13 (12.15%)	6 (5.71%)	9 (9.00%)	0 (0.00%)	1 (9.09%)	0 (0.00%)	0 (0.00%)
Paresthesia	3 (3.90%)	0 (0.00%)	0 (0.00%)	2 (2.00%)	1 (7.14%)	0 (0.00%)	0 (0.00%)	0 (0.00%)
Sore throat	11 (14.29%)	5 (4.67%)	6 (5.71%)	8 (8.00%)	0 (0.00%)	0 (0.00%)	0 (0.00%)	0 (0.00%)
Comorbidities								
Lung diseases	7 (9.09%)	11 (10.28%)	12 (11.43%)	21 (21.00%)	4 (28.57%)	3 (27.27%)	2 (8.70%)	2 (16.67%)
Cardiac diseases	15 (19.48%)	23 (21.50%)	54 (51.43%)	54 54.00%)	9 (64.29%)	6 (54.55%)	16 (69.57%)	10 (83.33%)
Dyslipidemia	3 (3.90%)	2 (1.87%)	11 (10.48%)	9 (9.00%)	2 (14.29%)	0 (0.00%)	0 (0.00%)	1 (83.33%)
Chronic Renal Failure	0 (0.00%)	1 (0.93%)	11 10.48%)	11 (11.00%)	2 (14.29%)	1 (9.09%)	6 (26.09%)	2 (16.67%)
Oncological diseases	3 (3.90%)	12 (11.21%)	13 (12.38%)	15 (15.00%)	1 (7.14%)	2 (18.18%)	9 (39.13%)	3 (25.00%)
Diabetes	2 (2.60%)	2 (1.87%)	19 (18.10%)	18 (18.00%)	3 (21.43%)	2 (18.18%)	2 (8.70%)	3 (25.00%)
Gastrointestinal diseases	9 (40.26%)	9 (8.41%)	11 (10.48%)	10 (10.00%)	4 (28.57%)	2 (18.18%)	4 (17.39%)	3 (25.00%)
Neurological or psychiatric diseases	6 (40.26%)	4 (3.74%)	14 (13.33%)	22 (22.00%)	3 (21.43%)	5 (45.45%)	8 (34.78%)	3 (25.00%)
Urologic diseases	5 (6.49%)	5 (4.67%)	9 (8.57%)	3 (3.00%)	3 (21.43%)	1 (9.09%)	6 (26.09%)	0 (0.00%)
Ophthalmological diseases	0 (0.00%)	1 (0.93%)	3 (2.86%)	3 (3.00%)	1 (7.14%)	0 (0.00%)	0 (0.00%)	0 (0.00%)
Immunological, rheumatological, or hematological diseases	7 (9.09%)	19 7.76%)	16 (15.24%)	25 (25.00%)	1 (7.14%)	0 (0.00%)	4 (17.39%)	2 (16.67%)
Ventilated patients	0 (0.00%)	0 (0.00%)	0 (0.00%)	0 (0.00%)	0 (0.00%)	0 (0.00%)	5 (21.74%)	0 (0.00%)

**Table 4 microorganisms-11-00925-t004:** ANOVA data of the analyzed blood parameters. Significant scores are shown in bold.

	Omicron COVID-19 Effect		
	dF	F-Value	*p*-Value
CK	3,425	2.74	**=0.042**
CK-MB	3,403	1.03	0.375
TnT	3,419	12.12	**<0.001**
FBG	3,422	36.03	**<0.001**
GGT	3,395	6.98	**<0.001**
Glycemia	3,427	18.59	**<0.001**
CRP	3,355	46.41	**<0.001**
INR	3,424	1.16	0.323
aPTT	3,422	8.08	**<0.001**
LDH	3,427	31.40	**<0.001**
Albumin	3,383	1.97	0.117
D-dimer	3,403	29.05	**<0.001**
MGB	3,275	8.60	**<0.001**
AST	3,427	2.76	**=0.041**
ALT	3,426	1.15	0.325
Ferritin sex sex *x* morbidity	3,384	32.27	**<0.001**
	1,384	15.99	**<0.001**
	3,384	9.35	**<0.001**

**Table 5 microorganisms-11-00925-t005:** AUC scores for the P/F ratio. CK, CK-MB, TNT, FBG, GGT, glycemia, CRP, INR, aPTT, LDH, albumin, D-dimer, MGB, AST, ALT, and ferritin (men and women—see Section 2). The highest scores (in bold) were disclosed for TNT, FBG, glycemia, CRP, and LDH, albumin, D-dimer, MGB, and ferritin in the *deceased* group, but only for FBG, D-dimer, and ferritin in the *ICU* group. Significant scores are shown in bold.

	*Deceased* vs. *Emergency*		*ICU* vs. *Emergency*	
	Area under the Curve (AUC)	95% CI for AUC	Area under the Curve (AUC)	95% CI for AUC
P/F ratio	0.792	0.695–0.889	0.294	0.189–0.399
CK	0.542	0.413–0.672	0.453	0.305–0.601
CK-MB	0.664	0.547–0.780	0.618	0.478–0.757
TnT	**0.920**	0.858–0.982	0.782	0.702–0.862
FBG	**0.844**	0.746–0.942	**0.852**	0.771–0.932
GGT	0.680	0.575–0.784	0.661	0.481–0.739
Glycemia	**0.805**	0.708–0.902	0.740	0.622–0.859
CRP	**0.925**	0.856–0.994	0.787	0.646–0.929
INR	0.653	0.540–0.765	0.504	0.379–0.629
aPTT	0.756	0.645–0.867	0.599	0.473–0.726
LDH	**0.871**	0.781–0.961	0.693	0.579–0.807
Albumin	**0.957**	0.922–0.993	0.764	0.635–0.893
D-dimer	**0.921**	0.881–0.961	**0.822**	0.731–0.913
MGB	**0.910**	0.833–0.986	0.756	0.637–0.876
AST	0.774	0.677–0.870	0.605	0.472–0.739
ALT	0.484	0.362–0.606	0.539	0.404–0.674
Ferritin	**0.948**	0.907–0.989	**0.812**	0.725–0.899
Ferritin (men)	**0.962**	0.924–1	0.585	0.411–0.759
Ferritin (women)	**0.956**	0.907–1	**0.953**	0.896–1

**Table 6 microorganisms-11-00925-t006:** Positive predictive values (PPV—probability that the patient has the condition when restricted to those patients who test positive) in the *deceased*, *ICU* and *hospital ward* groups and negative predictive values (NPV—probability that a patient who has a negative test result indeed does not have the condition) in the *emergency* group are based on the reference values (out of range for PPV; in range for NPV) for P/F ratio, CK, CK-MB, TNT, FBG, GGT, glycemia, CRP, INR, aPTT, LDH, albumin, D-dimer, MGB, AST, ALT, and ferritin (men and women—see Section 2). Significant scores are shown in bold.

	PPV (*Deceased*)	PPV (*ICU*)	PPV (*Hospital Ward*)	NPV (*Emergency*)
P/F ratio (≥400)	0.571	0.360	0.410	**0.908**
CK (20–200 U/L)	0.212	0.280	0.223	**0.916**
CK-MB (0–4.94 µg/L)	0.156	0.041	0.070	**0.965**
TnT (0–0.014 mg/L)	0.781	0.458	0.482	**0.893**
FBG (150–400 mg/dL)	**0.878**	**0.920**	0.176	**0.932**
GGT (8–61 U/L)	0.181	0.130	0.758	0.635
Glycemia (70–100 mg/dL)	**0.818**	0.720	0.548	0.657
CRP (0.1–6 mg/L)	0.785	0.333	0.294	**0.815**
INR (0.8–1.2)	0.182	0.120	0.584	**0.938**
aPTT (ratio 0.86–1.14)	0.531	0.160	0.362	0.785
LDH (135–225 U/L)	**0.878**	0.600	0.616	0.670
Albumin (3.5–5 g/dL)	0.625	0.181	0.196	**0.968**
D-dimer (50–420 µg/L)	**1**	**0.869**	0.796	0.565
MGB (28–72 µg/L)	0.591	0.318	0.586	0.304
AST (9–45 U/L)	0.242	0.160	0.141	**0.961**
ALT (10–40 U/L)	0.090	0.240	0.263	**0.837**
Ferritin (ng/mL)				
Ferritin (men 30–400)	**0.900**	0.250	0.515	0.800
Ferritin (women 15–150)	**0.909**	**0.899**	0.595	0.714

**Table 7 microorganisms-11-00925-t007:** Spearman Correlation values for the age parameter in the deceased group. Significant scores are shown in bold.

	Men			Women		
	SSD	Rho	*p*-Value	SSD	Rho	*p*-Value
P/F ratio	3108.50	−0.352	0.918	243.50	0.149	0.622
CK	1598.00	0.210	0.323	144.00	0.345	0.274
CK-MB	1052.00	0.406	0.062	128.00	0.224	0.501
TnT	349.00	0.803	**0.001**	243.00	−0.107	0.735
FBG	2040.00	−0.008	0.970	297.00	−0.350	0.268
GGT	2611.50	−0.290	0.173	174.00	−0.055	0.870
Glycemia	1739.00	0.141	0.509	268.00	−0.218	0.490
CRP	1009.50	0.114	0.627	102.00	0.382	0.252
INR	1732.50	0.144	0.499	214.00	0.027	0.931
aPTT	1198.00	0.324	0.138	196.00	0.109	0.730
LDH	2483.50	−0.227	0.286	136.50	0.380	0.230
Albumin	2018.00	0.003	0.988	268.50	−0.220	0.485
D-dimer	1123.00	0.366	0.093	102.00	0.382	0.252
MGB	193.00	0.470	0.103	146.00	0.115	0.729
AST	2631.00	−0.300	0.159	220.00	0.001	0.999
ALT	2722	−0.345	0.105	183.00	0.168	0.532
Ferritin	2306.50	−0.498	**0.025**	256.00	−0.164	0.604

**Table 8 microorganisms-11-00925-t008:** Vaccination effects on the single analyzed biomarkers in a two-way ANOVA. The sex effect was not considered because only the ferritin data differed between men and women (for ferritin, no effect was found for sex *x* vaccination). Significant scores are shown in bold.

Omicron COVID-19 and Vaccination Effects									
	Vaccination (Yes/No)			Omicron			Vaccination *x* Omicron		
	dF	F-Value	*p*-Value	dF	F-Value	*p*-Value	dF	F-Value	*p*-Value
P/F ratio	1,438	3.092	0.079	3	29.12	**0.001**	3	1.805	0.145
CK	1,423	0.731	0.393	3	2.741	**0.043**	3	1.247	0.292
CK-MB	1,401	0.154	0.695	3	0.760	0.517	3	0.199	0.897
TnT	1,417	3.927	**0.048**	3	9.233	**0.001**	3	1.378	0.249
FBG	1,420	1.302	0.255	3	37.44	**0.001**	3	4.042	**0.007**
GGT	1,393	1.250	0.264	3	8.478	**0.001**	3	3.261	**0.022**
Glycemia	1,425	2.844	0.092	3	14.90	**0.001**	3	2.089	0.101
CRP	1,353	1.368	0.243	3	52.06	**0.001**	3	0.891	0.446
INR	1,422	<0.001	1.000	3	1.526	0.207	3	0.301	0.825
aPTT	1,420	1.587	0.209	3	7.636	**0.001**	3	0.759	0.518
LDH	1,425	6.831	**0.009**	3	35.54	**0.001**	3	3.396	**0.018**
Albumin	1,381	0.085	0.771	3	1.637	0.180	3	0.261	0.853
D-dimer	1,401	0.010	0.919	3	28.99	**0.001**	3	3.758	**0.011**
MGB	1,274	1.864	0.173	3	6.686	**0.001**	3	0.767	0.513
AST	1,425	2.017	0.156	3	2.114	0.098	3	0.288	0.834
ALT	1,424	1.318	0.252	3	0.781	0.505	3	0.590	0.622
Ferritin	1,382	0.195	0.659	3	38.78	**0.001**	3	2.248	0.082

## Data Availability

Data are available upon request.

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
