# Peer review of "Blood Biomarkers from the Emergency Department Disclose Severe Omicron COVID-19-Associated Outcomes"

_microorganisms, 2023, doi:10.3390/microorganisms11040925_

Round 1
Reviewer 1 Report
In this single-center retrospective study in Italy, the study authors evaluated COVID-19 patients presenting during the Omicron wave and compared initial laboratory parameters in four different groups of patients. They noted that Troponin and Albumin were associated with poorer outcomes. Although the study raises certain interesting points that warrant further evaluation, there are certain points in the manuscript that must be addressed prior to considering for publication.
Major Comments
1. The study authors report on the vaccination status of their patient population in Table 1. While this is interesting information, how doe it correlate to their current study of evaluation of biomarkers? In the Discussion, they compare the mortality of vaccinated versus unvaccinated patients, but the correlation with biomarkers is not clear.
2. In Table 3, the study authors record the comorbidities of the patients in the study. However, details related to certain comorbidity groups is not explained – for example, what Oncological diseases were noted in the study? What Ophthalmological diseases were documented? If needed, it would be helpful to include this information in a Supplementary Table.
3. Adding to the above point, it is unclear whether there were significant differences at baseline between the groups between comorbidities. There are certain comorbidities (such as oncological or rheumatological diseases) that are known to have expected baseline elevation in inflammatory markers and also associated with worse outcomes in COVID-19. For example, oncological diseases were noted in 8.15% of group 1 and 34.3% of group 4 patients. It would be important to mention if these baseline comorbidities were statistically different and whether that reflected in the laboratory test results.
Minor Comments
1. Abstract line 23-24 (page 1 of 16) – the authors should consider changing “Actually, the Omicron strain…” to “Lately, the Omicron strain…..”.
2. Abstract line 26-27 (Page 1 of 16) – the authors should consider changing “However, only a few Omicron data are available” to “However, only few studies are available on early routine biochemical blood biomarkers for Omicron-afflicted patients” or something to that effect.
3. Page 2 of 16, line 51 – at beginning of the sentence, consider changing “Actually” to “Recently”.
4. Page 2 of 16, line 58 – consider changing “evidence, this, which contrasts” to “this contrasts sharply”
5. Page 3 of 16, lines 118-120 – please mention the testing platform on which RT-PCR testing was performed. Also mention if Cycle threshold values were collected.
6. Statistical Analysis (Page 5 of 16) – please mention what statistical software program was used in analysis.
Author Response
Reviewer 1
In this single-center retrospective study in Italy, the study authors evaluated COVID-19 patients presenting during the Omicron wave and compared initial laboratory parameters in four different groups of patients. They noted that Troponin and Albumin were associated with poorer outcomes. Although the study raises certain interesting points that warrant further evaluation, there are certain points in the manuscript that must be addressed prior to considering for publication.
Reply: we do thank the reviewer for the positive comments. As you will see, we replied point-by-point to each comment highlighting in light yellow the revised text.
Major Comments
- The study authors report on the vaccination status of their patient population in Table 1. While this is interesting information, how doe it correlate to their current study of evaluation of biomarkers? In the Discussion, they compare the mortality of vaccinated versus unvaccinated patients, but the correlation with biomarkers is not clear.
Reply: as suggested, we included further info and stat on the effects of vaccination on biomarkers and mortality. In particular, we added a Table (number 8), and 2 figures (number 8 and 9) to better clarify the data on vaccination. Accordingly, the discussion was modified (lines 383-410 of the revised paper).
- In Table 3, the study authors record the comorbidities of the patients in the study. However, details related to certain comorbidity groups is not explained – for example, what Oncological diseases were noted in the study? What Ophthalmological diseases were documented? If needed, it would be helpful to include this information in a Supplementary Table.
Reply: as requested, Table 3 was updated. The effect of comorbidities was analyzed in the Results (Figure 1, lines 191-205; lines 212-215) and Discussed (lines 411-420) in the revised paper.
- Adding to the above point, it is unclear whether there were significant differences at baseline between the groups between comorbidities. There are certain comorbidities (such as oncological or rheumatological diseases) that are known to have expected baseline elevation in inflammatory markers and also associated with worse outcomes in COVID-19. For example, oncological diseases were noted in 8.15% of group 1 and 34.3% of group 4 patients. It would be important to mention if these baseline comorbidities were statistically different and whether that reflected in the laboratory test results.
Reply: please see the previous reply.
Minor Comments
- Abstract line 23-24 (page 1 of 16) – the authors should consider changing “Actually, the Omicron strain…” to “Lately, the Omicron strain…..”.
Reply: as suggested, we modified the word (now line 23).
- Abstract line 26-27 (Page 1 of 16) – the authors should consider changing “However, only a few Omicron data are available” to “However, only few studies are available on early routine biochemical blood biomarkers for Omicron-afflicted patients” or something to that effect.
Reply: as suggested, we modified the sentence (now lines 25-26).
- Page 2 of 16, line 51 – at beginning of the sentence, consider changing “Actually” to “Recently”.
Reply: as suggested, we modified the word (now line 51).
- Page 2 of 16, line 58 – consider changing “evidence, this, which contrasts” to “this contrasts sharply”
Reply: as suggested, we modified the sentence (now line 65).
- Page 3 of 16, lines 118-120 – please mention the testing platform on which RT-PCR testing was performed. Also mention if Cycle threshold values were collected.
Reply: as requested, we provided more info (now lines 125-128).
- Statistical Analysis (Page 5 of 16) – please mention what statistical software program was used in analysis.
Reply: as requested, we provided the appropriate info in the revised Statistical analysis section (now lines 181-189).
Reviewer 2 Report
In this manuscript, Pennacchia et al. describe blood biomarkers to predict the severity of outcomes associated with COVID-19 caused by infection with the SARS-CoV-2 Omicron lineage. The topic is interesting since a small proportion of individuals still undergo hospitalization or die from COVID-19 even after vaccination. Therefore, it is important to determine factors to predict severity in COVID-19 and to define aspects such as comorbitities still contributing for severity in a context of immunized population and the circulation of Omicron variant.
Please find below a few considerations on this paper.
Throughout the Introduction, many known biomarkers for severe COVID-19 were cited. However, the complications associated with hospitalization, such as coagulopathies and the cytokine storm-induced general inflammation, were not presented. The clinical picture of COVID-19 would therefore need some attention. In addition, the radical drop in the numbers of hospitalization and death after the introduction of mass vaccination, which changed the scenario of the pandemic, was ignored in the introduction. The search for biomarkers for severe COVID-19 in individuals after vaccination should be also highlighted in the introduction, since the authors considered this variable into the definition of the study population. Indeed, the part describing the separation of the study population into four groups in the last paragraph of the introduction (lines 88-94) could be excluded since it is shown in details in the Methods section.
In the unvaccinated group, immunity against SARS-CoV-2 infection was not ruled out by checking for previous infection. If this data is not available, the screening for anti-SARS-CoV-2 nucleoprotein IgG could be performed if only mRNA and adenovirus-based vaccines but not inactivated virus vaccines were not used in Italy. If the recovery of serum samples is not possible, the inability to identify individuals that were previously infected should be highlighted as a limitation of the study.
In the description of the statistical analysis (lines 171-174), it is stated that "data were analyzed to assess normality by Pearson's chi-squared test and two-way analysis of variance (ANOVA) (emergency vs ICU vs deceased and males vs females) was used to analyze the laboratory parameters. This sentence is confusing. Please rephrase it to explain the methods accordingly.
In Table 3, please provide percentage values inside parenthesis next to the numbers to facilitate the comparison between groups.
Data shown in Table 3 was not explored by comparative/descriptive analysis, particularly considering preexisting clinical conditions that can act as comorbidities for COVID-19. The frequencies of such conditions according to the outcomes should be analyzed and taken into consideration, and also discussed.
In the Figures 1, 2, 3, 4, 5, and 6, it is suggested to organize the groups according to the clinical severity (emergency < hospital ward < ICU < deceased) to have a better view of the data.
In the description of the Results, please substitute the term "gender" for "sex" along the text and tables. According to the WHO definition, gender is a social construction and sex refers to biological and physiological characteristics (https://www.who.int/health-topics/gender#tab=tab_1).
Line 210: It was indicated that "Blood parameters are shown in Figure 2, 3, 4, 5 and 6". Please exclude number 2 from the sentence since data from Figure 2 was already described and the blood parameters presented after do not include the Horowitz index.
Line 221: The sentence "Table 5. and 6 show the ROC data." seems lost in the text.
In general, please make reference to Tables and Figures after data description for a better comprehension of the Results presentation.
Lines 295-299: the effectiveness of vaccination against death by COVID-19 was discussed in light of differences between groups according to sex, but the group sizes are too small to have a conclusion. This seems a size effect rather than a real difference. Therefore, please include a reference in this paragraph on the effectiveness of vaccination against death by COVID-19 among males and females to support your point, or please delete the sentence (lines 295-297) "Other interesting findings of the present investigation are the contradictory vaccination data showing the highest percentage of unvaccinated women in the deceased group but also a high percentage in vaccinated men", since this would mislead readers.
In the discussion (lines 305-312), many of the findings are discussed considering the association only within male or female sex. However, data presented in the Results section was not shown considering sex. Please adjust the data presentation in the Results section to the findings discussed.
Recently, Park et al. published changes in blood biomarkers according to SARS-CoV-2 lineages. This may help to enrich the discussion.
Park, C.; Tavakoli-Tabasi, S.; Sharafkhaneh, A.; Seligman, B.J.; Hicken, B.; Amos, C.I.; Chou, A.; Razjouyan, J. Inflammatory Biomarkers Differ among Hospitalized Veterans Infected with Alpha, Delta, and Omicron SARS-CoV-2 Variants. Int. J. Environ. Res. Public Health 2023, 20, 2987. https://doi.org/10.3390/ijerph20042987.
There is no discussion on the impact of comorbidities on the outcome of Omicron COVID-19. Since liver, cardiac, inflammatory, and coagulation biomarkers were evaluated, alterations should also consider preexinting diseases.
The text has several misused words and unusual sentence constructions. It is recommended to have the manuscript read and revised by a native English speaker.
Minor changes:
In the Abstract, please provide the names of the factors instead of abbreviations for: TnT, FBG, CPR, LDH, MGB.
Author Response
Reviewer 2
In this manuscript, Pennacchia et al. describe blood biomarkers to predict the severity of outcomes associated with COVID-19 caused by infection with the SARS-CoV-2 Omicron lineage. The topic is interesting since a small proportion of individuals still undergo hospitalization or die from COVID-19 even after vaccination. Therefore, it is important to determine factors to predict severity in COVID-19 and to define aspects such as comorbitities still contributing for severity in a context of immunized population and the circulation of Omicron variant.
Reply: we do thank the reviewer for the positive comments. As you will see, we replied point-by-point to each comment highlighting in light yellow the revised text.
Please find below a few considerations on this paper.
Throughout the Introduction, many known biomarkers for severe COVID-19 were cited. However, the complications associated with hospitalization, such as coagulopathies and the cytokine storm-induced general inflammation, were not presented.
Reply: as suggested, we included further info on the complications associated with hospitalization (now lines 56-62).
The clinical picture of COVID-19 would therefore need some attention. In addition, the radical drop in the numbers of hospitalization and death after the introduction of mass vaccination, which changed the scenario of the pandemic, was ignored in the introduction.
Reply: according to this comment, we included further info on the role of mass vaccination in COVID-19 pandemic control (now lines 66-69).
The search for biomarkers for severe COVID-19 in individuals after vaccination should be also highlighted in the introduction, since the authors considered this variable into the definition of the study population.
Reply: as requested, we highlighted in the introduction the importance of searching biomarkers for severe COVID-19 in individuals after vaccination (now lines 74-77).
Indeed, the part describing the separation of the study population into four groups in the last paragraph of the introduction (lines 88-94) could be excluded since it is shown in details in the Methods section.
Reply: as suggested, this sentence was removed.
In the unvaccinated group, immunity against SARS-CoV-2 infection was not ruled out by checking for previous infection. If this data is not available, the screening for anti-SARS-CoV-2 nucleoprotein IgG could be performed if only mRNA and adenovirus-based vaccines but not inactivated virus vaccines were not used in Italy. If the recovery of serum samples is not possible, the inability to identify individuals that were previously infected should be highlighted as a limitation of the study.
Reply: we do thank the reviewer for raising this key point. Accordingly, we included in the study limitations this issue (now lines 481-491).
In the description of the statistical analysis (lines 171-174), it is stated that "data were analyzed to assess normality by Pearson's chi-squared test and two-way analysis of variance (ANOVA) (emergency vs ICU vs deceased and males vs females) was used to analyze the laboratory parameters. This sentence is confusing. Please rephrase it to explain the methods accordingly.
Reply: as suggested, the paragraph was rewritten (now lines 181-189).
In Table 3, please provide percentage values inside parenthesis next to the numbers to facilitate the comparison between groups.
Reply: as suggested, the table was updated.
Data shown in Table 3 was not explored by comparative/descriptive analysis, particularly considering preexisting clinical conditions that can act as comorbidities for COVID-19. The frequencies of such conditions according to the outcomes should be analyzed and taken into consideration, and also discussed.
Reply: according to the comment of the reviewer, we included in the revised paper additional analyses on the effects of comorbidities and vaccination on the affected individuals. In particular, we added a Table (number 8), and 3 figures (number 1, 8 and 9) to better clarify the data on vaccination. Accordingly, the Result section and discussion were modified (lines 191-205, lines 212-215, lines 295-306, and lines 383-419 of the revised paper).
In the Figures 1, 2, 3, 4, 5, and 6, it is suggested to organize the groups according to the clinical severity (emergency < hospital ward < ICU < deceased) to have a better view of the data.
Reply: we drew the figures ordering the groups (deceased<emergency<ICU< hospital ward) on purpose to compare putative differences due to mortality or severe COVID-19 with the “controls” (the emergency group). However, we may redraw the figures if necessary.
In the description of the Results, please substitute the term "gender" for "sex" along the text and tables. According to the WHO definition, gender is a social construction and sex refers to biological and physiological characteristics (https://www.who.int/health-topics/gender#tab=tab_1).
Reply: we apologize for the misunderstanding. As suggested, we replaced the term gender with sex.
Line 210: It was indicated that "Blood parameters are shown in Figure 2, 3, 4, 5 and 6". Please exclude number 2 from the sentence since data from Figure 2 was already described and the blood parameters presented after do not include the Horowitz index.
Reply: we do thank the reviewer for the comment. Accordingly, appropriate changes were made (now lines 247-249).
Line 221: The sentence "Table 5. and 6 show the ROC data." seems lost in the text.
Reply: we again apologize for the inappropriate formatting. Accordingly, the sentence was moved on line 259.
In general, please make reference to Tables and Figures after data description for a better comprehension of the Results presentation.
Reply: in the original version we carefully followed in the template the results’ MDPI rules. Furthermore, some tables and figures have been moved by the MDPI editorial service after the submission. However, we formatted again the paper for a better comprehension of data, tables and figures.
Lines 295-299: the effectiveness of vaccination against death by COVID-19 was discussed in light of differences between groups according to sex, but the group sizes are too small to have a conclusion. This seems a size effect rather than a real difference. Therefore, please include a reference in this paragraph on the effectiveness of vaccination against death by COVID-19 among males and females to support your point, or please delete the sentence (lines 295-297) "Other interesting findings of the present investigation are the contradictory vaccination data showing the highest percentage of unvaccinated women in the deceased group but also a high percentage in vaccinated men", since this would mislead readers.
Reply: as suggested, the paragraph was totally updated and rewritten (now lines 383-410).
In the discussion (lines 305-312), many of the findings are discussed considering the association only within male or female sex. However, data presented in the Results section was not shown considering sex. Please adjust the data presentation in the Results section to the findings discussed.
Reply: as shown in Table 4, no sex effects were found and these data were not shown (except for Ferritin).
Recently, Park et al. published changes in blood biomarkers according to SARS-CoV-2 lineages. This may help to enrich the discussion.
Park, C.; Tavakoli-Tabasi, S.; Sharafkhaneh, A.; Seligman, B.J.; Hicken, B.; Amos, C.I.; Chou, A.; Razjouyan, J. Inflammatory Biomarkers Differ among Hospitalized Veterans Infected with Alpha, Delta, and Omicron SARS-CoV-2 Variants. Int. J. Environ. Res. Public Health 2023, 20, 2987. https://doi.org/10.3390/ijerph20042987.
Reply: as suggested, the data of this recent publication were discussed in the revised paper (now lines 456-464).
There is no discussion on the impact of comorbidities on the outcome of Omicron COVID-19. Since liver, cardiac, inflammatory, and coagulation biomarkers were evaluated, alterations should also consider preexinting diseases.
Reply: as suggested by both reviewers, additional analyses on the impact of comorbidities on the outcome of Omicron COVID-19 were provided in the revised version of the paper. Indeed, Table 3 was updated. The effect of comorbidities was analyzed in the Results (Figure 1, lines 191-205, lines 212-215) and Discussed (lines 411-420) in the revised paper.
The text has several misused words and unusual sentence constructions. It is recommended to have the manuscript read and revised by a native English speaker.
Reply: we apologize for these misused words. We made changes throughout the text.
Minor changes:
In the Abstract, please provide the names of the factors instead of abbreviations for: TnT, FBG, CPR, LDH, MGB.
Reply: as suggested, in the Abstract these abbreviations were written in full (now lines 34-37).
Round 2
Reviewer 1 Report
The authors have addressed the previously raised queries. I have no other comments at this time.
Author Response
The authors have addressed the previously raised queries. I have no other comments at this time.
Reply: we thank the reviewer.
Reviewer 2 Report
In this manuscript, Pennacchia et al. describe blood biomarkers to predict the severity of outcomes associated with COVID-19 caused by infection with the SARS-CoV-2 Omicron lineage. The topic is interesting since a small proportion of individuals still undergo hospitalization or die from COVID-19 even after vaccination. Therefore, it is important to determine factors to predict severity in COVID-19 and to define aspects such as comorbitities still contributing for severity in a context of immunized population and the circulation of Omicron variant.
The authors have answered all questions and have also made substantial modifications in this revised version of the paper.
However, a few modifications are suggested below to improve the text comprehension by readers.
1) Lines 202-205: "Furthermore, ANOVA also revealed an effect of vaccination (see Figure 1, right panel) on concomitant comorbidities with an elevation in vaccinated individuals [F(1,441)=4.40, p<0.05, see post-hoc comparisons in Figure 1, right panel] particularly evident for the hospital ward group, again in the absence of a sex effect."
The sentence seems odd. Data presented shows that there is a higher frequency of comorbidities among vaccinated individuals compared to unvaccinated patients.
As a suggestion, please rephrase the sentence to: Furthermore, ANOVA also revealed a higher frequency of comorbidities among vaccinated compared to unvaccinated patients [F(1,441)=4.40, p<0.05, see post-hoc comparisons in Figure 1, right panel], particularly evident for the hospital ward group, again in the absence of a sex effect.
2) Lines 387-388: The authors discuss that "Surprisingly, vaccinated patients had higher TnT values when compared to unvaccinated individuals. However, the group sizes are too small to draw a conclusion".
Was there any difference in the frequency of cardiac diseases among vaccinated and unvaccinated patients? If not, please state that differences in TnT levels were not associated with the frequency of preexisting cardiac diseases.
3) In the Abstract, it is concluded that "Compared to previous Delta COVID-19 parallel emergency patterns of prediction, Omicron-induced changes in TnT and Albumin may be considered other early predictors of severe outcomes."
Although it is clearly presented along the paper that TnT levels are distinct among groups afflicted by Omicron when discriminated by clinical severity and other characteristics, it is not clear why Albumin was also included in the conclusion as an early predictor, since alterations in Omicron cases were not consistent. There is only a sentence in the discussion (lines 436-439) with a self-citation highlighting differences to findings from a previous study during the Delta wave of COVID-19.
Please highlight findings along the text that support this conclusion and include more information in the discussion.
Author Response
Reviewer 2
In this manuscript, Pennacchia et al. describe blood biomarkers to predict the severity of outcomes associated with COVID-19 caused by infection with the SARS-CoV-2 Omicron lineage. The topic is interesting since a small proportion of individuals still undergo hospitalization or die from COVID-19 even after vaccination. Therefore, it is important to determine factors to predict severity in COVID-19 and to define aspects such as comorbitities still contributing for severity in a context of immunized population and the circulation of Omicron variant.
The authors have answered all questions and have also made substantial modifications in this revised version of the paper.
Reply: we do thank the reviewer for her/his comments
However, a few modifications are suggested below to improve the text comprehension by readers.
1) Lines 202-205: "Furthermore, ANOVA also revealed an effect of vaccination (see Figure 1, right panel) on concomitant comorbidities with an elevation in vaccinated individuals [F(1,441)=4.40, p<0.05, see post-hoc comparisons in Figure 1, right panel] particularly evident for the hospital ward group, again in the absence of a sex effect."
The sentence seems odd. Data presented shows that there is a higher frequency of comorbidities among vaccinated individuals compared to unvaccinated patients.
As a suggestion, please rephrase the sentence to: Furthermore, ANOVA also revealed a higher frequency of comorbidities among vaccinated compared to unvaccinated patients [F(1,441)=4.40, p<0.05, see post-hoc comparisons in Figure 1, right panel], particularly evident for the hospital ward group, again in the absence of a sex effect.
Reply: we sincerely thank the reviewer for the suggestion. Accordingly, we replaced the sentence (lines 202-205).
2) Lines 387-388: The authors discuss that "Surprisingly, vaccinated patients had higher TnT values when compared to unvaccinated individuals. However, the group sizes are too small to draw a conclusion".
Was there any difference in the frequency of cardiac diseases among vaccinated and unvaccinated patients? If not, please state that differences in TnT levels were not associated with the frequency of preexisting cardiac diseases.
Reply: as requested, we included further info (lines 388-390).
3) In the Abstract, it is concluded that "Compared to previous Delta COVID-19 parallel emergency patterns of prediction, Omicron-induced changes in TnT and Albumin may be considered other early predictors of severe outcomes."
Although it is clearly presented along the paper that TnT levels are distinct among groups afflicted by Omicron when discriminated by clinical severity and other characteristics, it is not clear why Albumin was also included in the conclusion as an early predictor, since alterations in Omicron cases were not consistent. There is only a sentence in the discussion (lines 436-439) with a self-citation highlighting differences to findings from a previous study during the Delta wave of COVID-19.
Please highlight findings along the text that support this conclusion and include more information in the discussion.
Reply: we apologize for the typing mistake. We revised the text by removing Albumin (lines 37-39; lines 438-442).